Evaluation of NEUROG1 methylation status in stool specimens in the advanced adenomas and colorectal cancer

Zhang Lijin 1 2 3
Lin Aiping 1 2 3
Lin Jie 4
Chen Juan 1 2 3
Chen Mengshi 1 2 3
Yu Xunbin 4
Wu Yijuan 4
Wang Tao 5
Lu Yan 5
Ba Zhaofen 5
Cheng Xiaowei 5
Zhang Tiantian 5
Chen Minghong mhchen9035@sohu.com 1 2 3
1 Department of Gastroenterology, the Shengli Clinical Medical College, Fujian Medical University , Fuzhou , Fujian , China
2 Department of Gastroenterology, Fujian Provincial Hospital , Fuzhou , Fujian , China
3 Department of Gastroenterology, Fuzhou University Affiliated Provincial Hospital , Fuzhou , Fujian , China
4 Department of Pathology, Fujian Provincial Hospital , Fuzhou , Fujian , China
5 Jiangsu Microdiag Biomedical Technology Co., LTD , Suzhou , Jiangsu , China
Uversky Vladimir
Electronic publication date: 2025 Oct 8
Publication date: 2025
Volume: 13
Electronic Location ID: e19968
Received 2024 Nov 22; Accepted 2025 Jul 31
Copyright: ©2025 Zhang et al.
Copyright year: 2025
Copyright holder: Zhang et al.
License: This is an open access article distributed under the terms of the Creative Commons Attribution License, which permits unrestricted use, distribution, reproduction and adaptation in any medium and for any purpose provided that it is properly attributed. For attribution, the original author(s), title, publication source (PeerJ) and either DOI or URL of the article must be cited.
License URL: https://creativecommons.org/licenses/by/4.0/

Keywords: Colorectal cancer, NEUROG1, Stool, DNA methylation, Diagnosis

Funding: Natural Science Foundation of Fujian Province, China 2021J01367 This study was supported by the Natural Science Foundation of Fujian Province, China (Grant No. 2021J01367). The funders had no role in study design, data collection and analysis, decision to publish, or preparation of the manuscript.

==============================
Background

To assess the status of NEUROG1 methylation in the advanced adenoma and colorectal cancer.

Methods

The NEUROG1 methylation in tissue and stool samples from patients with colorectal cancer (CRC), advanced adenoma (AA), and non-advanced adenoma (NAA) were evaluated using methylation-specific quantitative polymerase chain reaction (PCR).

Results

In tissue samples, the NEUROG1 methylation detection rates were 36% for CRC, 24% for NAA, and 88% for AA. In stool samples, the NEUROG1 methylation detection had a sensitivity of 63.46% for CRC with a positive predictive value (PPV) of 85.94%. The overall diagnostic specificity of the test for the NAA and the healthy control was 76.32%, with a negative predictive value (NPV) of 40.28%.

Conclusion

NEUROG1 methylation detection can potentially be used in the CRC and AA screening.

Introduction

Colorectal cancer (CRC) is one of the leading causes of cancer-related death worldwide. According to the latest statistics, there are approximately 104,270 new cases of CRC in the United States annually, making it the third most common form of cancer in that country. In addition, the mortality rate is the second-highest among all cancer-related deaths in the United States (Siegel & Miller, 2021). In China, the incidence of CRC ranks fourth after lung cancer, breast cancer, and stomach cancer, and mortality is also fourth among all cancers (Du et al., 2017). Furthermore, the prognosis is poor for patients with advanced metastatic CRC, which has a five-year survival rate of less than 10%. However, most patients with CRC can benefit from surgery when diagnosed early, and the five-year survival rate of Tumor, Node, Metastasis (TNM) stage 1 CRC is greater than 90% (Werner et al., 2016). Therefore, the early screening and diagnosis of CRC play a critical role in positive clinical outcomes. Recently, several countries have initiated CRC screening programs, and a series of consensuses were published on the early screening guidelines of CRC (2021).

The most common methods for CRC screening include colonoscopy, fecal occult blood testing (FOBT) and fecal immunochemical test (FIT). FOBT comprises hydrogen peroxide for detecting fecal occult blood test, known as the guaiac-based FOBt (gFOBt). The fecal immunochemical test (FIT) is based on immunochemistry. FIT has had a distinct advantage over gFOBt, its lack of dietary restrictions prior to sample collection (Benton, Seaman & Halloran, 2015). FIT has gradually replaced gFOBt in the clinic, but studies have indicated its effectiveness is relatively limited to early-stage CRC diagnosis (Tepus & Yau, 2020). Hence, there is an urgent need for newer screening systems for early-stage CRC.

Colonoscopy is the “gold standard” diagnostic method for CRC due to its high sensitivity and specificity; however, it is invasive, requires skilled technical expertise, and patient noncompliance (Ziegler et al., 2010).

Accumulated evidence has indicated that CRC is a genetic-driven disease driven by DNA mutations, chromatin abnormalities, and epigenetic changes that influence the expression of critical oncogenes (Dickinson et al., 2015; Huang et al., 2018). Epigenetic changes, such as abnormal DNA methylation, non-coding RNA (miRNA and siRNA), and histone modifications, are closely associated with CRC development and progression (Okugawa, Grady & Goel, 2015).

Abnormal DNA hypomethylation in the promoter regions of tumor suppressor genes was reported to be an early event in CRC. For example, DNA hypomethylation of the promoter region of Secreted frizzled-related protein 2, SFRP2 gene activates the Wnt signalling pathway and promotes tumorigenesis in CRC (Zhang et al., 2014).

Studies have shown that genetic methylation biomarkers can be detected in body fluids, such as blood, urine, and stool. Hence, they may serve as novel biomarkers for CRC screening in the future. Furthermore, stool samples from patients with cancer often contain more DNA than blood, as tumor cells can be shed off from mass and excreted through the stool. Studies have shown that the sensitivity of stool samples is significantly higher than plasma samples (Ahlquist et al., 2012a).

In the last decade, stool DNA methylation detection has become a non-invasive, highly specific, and cheap screening method for diagnosing CRC. In 2014, the United States Food and Drug Administration (FDA) approved a multi-target stool DNA (MT-sDNA) test for screening CRC among high-risk asymptomatic patients (Imperiale et al., 2014).

Recently, it was demonstrated that abnormal methylation of genes adenomatous polyposis coli (APC), ataxia telangiectasia mutated (ATM), cyclin-dependent kinase inhibitor 2A (CDKN2A), GATA binding protein 4 (GATA4), and secreted frizzled-related protein 2 (SFRP2) could be used for CRC screening in stool samples (Kadiyska & Nossikoff, 2015; Laugsand, Brenne & Skorpen, 2021).

Neurogenin 1 (NEUROG1) is one of the classic methylation biomarkers that can distinguish the degree of the CpG island methylator phenotype (CIMP) (Ibrahim et al., 2011). Previous studies have shown that the methylation of NEUROG1 could be used as a serum biomarker for early-stage CRC (Herbst et al., 2011; Otero-Estévez & Gallardo-Gomez, 2020). However, it has not been investigated for stool samples.

This study aimed to evaluate the status of the NEUROG1 methylation in the AA and CRC. Herein, we detected NEUROG1 methylation levels in stool samples and tumor specimens from CRC patients.

Materials and Methods

Patients and sample selection

The study protocol is shown in Fig. 1, the formalin-fixed, paraffin-embedded (FFPE) samples were collected and stored at room temperature from a total of 75 patients, 25 with CRC, 25 with non-advanced adenomas (NAA), and 25 with advanced adenomas (AA) were diagnosed at the First Affiliated Hospital of Henan University of Science and Technology from May 2019 to May 2020. The patients were confirmed by two experienced doctors, depending on the colonoscopy results and along with the pathology results of CRC, AA, and NAA.

Figure 1 Experimental design in this study.

Two separated cohorts were included: the FFPE cohort (N = 75) and the stool cohort (N = 272). NEUROG1 methylation detection and analysis were performed on all the samples enrolled.

Stool samples were collected from 272 patients diagnosed and treated at Fujian Provincial Hospital from July 2019 to December 2023, including 104 CRC patients, 92 AA patients, and 39 NAA patients. None of the patients had received any anticancer treatment before admission. Control stool samples were collected from 37 healthy individuals undergoing colonoscopy, and CRC, AA, NAA patients were excluded.

In addition, several clinical characteristics were collected, including age, sex, and classification.

This study was approved by the Institutional Review Board of Fujian Provincial Hospital (K2019-11-027). Informed consent was obtained from all enrolled patients and healthy control subjects.

Stool samples were collected before tumor resection and stored at −80 °C in a storage buffer.

DNA extraction from stool

Stool DNA was extracted according to the operation manual and published protocol using a DNA extraction kit in the stool (Ahlquist et al., 2012b). After thawing, the buffered stool samples were homogenized with an oscillator and centrifuged. Then, an aliquot of 12 mL stool supernatant was treated with 50 mg/mL polyvinylpoly pyrrolidone PVPP (Aladdin, Shanghai, China).

The target gene sequence was directly captured by hybridization with the oligonucleotide probe (CGTGCAGCGCCCGGGTATTTGCATAATTTATGCTCGCGGGAGGCCGCCATCGCCCCTCCCCCAACCCGGAGTGTGCCCGTAATTACCG).

For this purpose, 10 mL polyvinylpolypyrrolidone-treated supernatant was denatured using 2.4 M (molar) guanidine isothiocyanate at 92 °C for 15 min (Aladdin, Shanghai, China). Next, 50 µL oligonucleotide capture probe-modified carboxyl magnetic beads (JSR) were added, mixed, and incubated at room temperature for 30 min.

The beads were collected using the magnetic rack and washed with washing buffer (10 mM MOPS, 150 mM NaCl, pH 7.5) three times. Finally, 50 µL nuclease-free water containing 20 ng/µL transfer RNA (Merck K GaA, Darmstadt, Germany) was added to the eluted DNA.

The 50 µL elution was used for bisulfite conversion, and the purified DNA was eluted to 60 µL in the elution buffer. The bisulfite transformation Kit (Zymo Research, Irvine, CA, USA) was used for DNA bisulfite transformation and purification of transformation products. All steps were performed following the manufacturer’s instructions.

DNA extraction of FFPE specimens

TIANamp FFPE DNA Kit (Tiangen Biotech Co., Ltd, Beijing, China) was used to isolate DNA from FFPE tissue samples. Briefly, 4–5 sections of FFPE specimens were collected in 1.5 mL centrifuge tubes. Deparaffinization and dehydration were performed by adding dimethylbenzene and 100% ethanol. After air drying, 400 µL digest buffer with 20 µL proteinase K was added to the precipitation. Then, the precipitation was suspended and digested at 55 °C for one hour. Following a one-hour incubation at 90 °C, the suspension was added into spin columns. After cleaning and centrifugation, the spin columns were dried and eluted with 50–100 µL elution buffer. The DNA samples were stored at −20 °C.

After extraction, the concentration of DNA was determined with the Qubit 2.0 Fluorometer (Thermo Fisher Scientific Inc., California, USA). One µg of extracted DNA was treated with bisulfite using the EZ DNA Methylation-Gold Kit (Zymo Research, Irvine, CA, USA).

Methylation-specific Quantitative PCR (MSP)

The methylation status of NEUROG1 was evaluated by Methylation-specific quantitative PCR (MSP) (Suzhou MicroDiag Biomedicine Co., Ltd, Suzhou, China). Sequences of the primers and probes for NEUROG1 and β-Actin (ACTB) were designed and synthesized by GENEWIZ, (Suzhou, China). The ACTB gene, located on chromosome 7 (7p22.1), was used as a reference. The sequences of the primers and probes for the indicated genes are as follows: NEUROG1_Forward: TCGTGTAGCGTTCGGGT, NEUROG1_Reverse: CACTCCGAATTAAAAAAAAAACG, NEUROG1_probe: ATCCCGCGAACATA; the sequences of predicted promoters and amplified regions in this study are as follows: the purple font is the predicted promoter, and the blue font is the amplified region in this study.

The amplified region is upstream from 172 to 243 of the promoter, i.e., −172 to −243

>hg19_dna range=chr5:134871000-134871843 5′pad=0 3′pad=0 strand=+ repeatMasking=none

CTTGGTGTCGTCGGGGAACGAGGGCAGCACGCTGCGCAGTGCGTCCAGGGCC GCGTTCAAGTTGTGCATGCGGTTGCGCTCGCGATCGTTGGCCTTGACGCGCCGG CTCCTGCGCAGCGAGTGCAGCAGCGCCTCGGAGCGGACCCGCGTCCGGCCG CGGCGCCGCCGCCTCTCCTGCTCGTCGTCCTGTGCCCCTGGAACCTCAGACG CCCGGGAGATATTGGGCGCGCCCCTGCGGGCCGGCGCGGGCGGCCCCGAAGCG GAGGCTGCCTGTTGGAGTCTGGCACAGTCTTCCTCGTCGGTGAGGAAGCCGGAT AGGTCACTGCCGCTGCTGCTGGCGCAGTCGAGGTCGGAGATGCAGGTCTCAAGGC GGGCTGGCATCGTTGCGCTGTGCAGGACCGACGGACAGATAGAAAGGCGCTCAGA GCGCTGCAGCCCGGACTGAGGGCAGAGCCGCCAGGGCGCACTTACGTTCCCAA CAGCCTGGGGTTGTTACTCTGTGCCAGTTGCGGGTGCGAGAGCCTGGAAGGG TGCAGGGGCGCACGGAGAACTTGGCCTGGCCTCCTCGCCTCGCCTGCAGGGG CCACGCGCCCGGCCGGTCTCCTGAGTGATGTCGCCGGCGATCAGATCAGCTCGTG TGAGCACCGAGTGTGGCACACGACTGGCCTCAGGACCCCTTAAGTACCCGGCGC AACAATGGGCGCCCCCCTCCCTTGCCACCTCCGCCCCCGCGGCAGCCCGGGTGA ATGGAGCGAGGCGGCAGGTCATCCCCGTGCAGCGCCCGGGTATTTGCATAATTT ATGCTCGCGGGAGGCCGCCATCGCCCCTCCCCCAACCCGGAGTG ACTB_Forward: GTGATGGAGGAGGTTTAGTAAGTT, ACTB_Reverse: CCAATAAAACCTACTCCTCCCTTAA, ACTB_probe: ACCACCACCCAACACACAATAACAAACACA. In brief, the total volume of qPCR was 30 µL, including 15 µL of DNA sample and 15 µL of PCR master mix. Next, qPCR was conducted on a LightCycler 480 II thermal cycler (Roche Diagnostics Corporation, Indianapolis, USA) using the following conditions: denaturation 95 °C for 20 min one cycle; 45 cycles (annealing 95 °C for 20 s and extension 60 °C for 35 s; and finally chilled to 40 °C for 30 s). After the amplification, the data were analyzed on LC480-II (Roche Diagnostics Corporation, Indianapolis, USA).

A standard curve was elaborated for NEUROG1 methylation to calibrate the investigation system (100%–0% bisulfite-converted NEUROG1-positive genomic DNA; slope = −0.2762; R2 = 0.9906). In brief, bisulfite conversion was performed of genomic DNA extracted from cells, in which the NEUROG1 promoter region was fully methylated (HCT116 cells) or unmethylated (293T cells) using a commercially available kit (EZ DNA Methylation-Gold Kit, Enzo, USA). Then, bisulfite-converted fully methylated genomic DNA was mixed with bisulfite-converted unmethylated DNA to obtain a bisDNA gradients (where NEUROG1 methylated DNA was 100%, 50%, 25%, 10%, 5%, 1%, 0.5%, 0.1%, 0.05%, and 0% to the total mixture), while the whole mixture’s bisDNA concentration was maintained at two ng/µL. After that, the samples were subjected to amplification and the results were plotted to generate the standard curve. Negative and positive cell DNA were purchased from Fubio Biotechnology, Co., Ltd, (Suzhou, China).

The quality control’s cycle threshold/crossing point (CP) value is 27–30 for tissue methylation DNA. For stool DNA detection, the CP value of the quality control needs to be <36. For samples without amplification in the target channels, 45 (maximum amplification cycle number of PCR) should be taken for the fitting calculation.

SPSS software (IBM Corp., Armonk, NY, USA) was used to conduct the fitting logistic regression analysis of CP values for target genes and internal reference genes, and the fitting formula was obtained: 11+e−7.867−0.097∗NEUROG1−0.098∗ACTB.

The sample is positive if the fitting value is > 0.7692.

Statistical analysis

Statistical analysis and receiver operating characteristic (ROC) curves were performed using SPSS software version 22.0 (IBM Corp., Armonk, NY, USA). The cut-off value was determined by using the maximum principle of the Youden’s index. The detection performance of stool NEUROG1 gene methylation in CRC and precancerous lesions was investigated. All the data were presented in percentages (%). The Chi-square test was used to compare the data, and P-values <0.05 were considered statistically significant.

Results

NEUROG1 methylation detection in FFPE specimens

To elucidate the methylation status of the promoter region of NEUROG1 in colon cancer, MSP was conducted in both FFPE specimens and patients’ feces samples. We determined the detection rates of NEUROG1 methylation in 75 FFPE samples, including (25 CRC, 25 AA, and 25 NAA) as shown in (Table 1). The methylation rates for CRC was 36%, and for NAA was 24%. Notably, the methylation level of AA was 88% and exhibited positive NEUROG1 methylation. NEUROG1 methylation detection rate was significantly different between CRC+AA group and NAA group in FFPE specimens, as shown in (Table 2).

Table 1 Detection rates of NEUROG1 methylation in FFPE specimens.

Subgroup	Number	Detection number	
CRC	25	9 (36%)	
AA	25	22 (88%)	
NAA	25	6 (24%)	
Notes.

CRC colorectal cancer

AA advanced adenoma

NAA non-advanced adenoma

Table 2 Comparation the rates of NEUROG1 methylation between CRC+AA group and NAA group in FFPE specimens.

Subgroup	Number	Positive	Negative	
CRC+AA	50	31	19	
NAA	25	6	19	
Notes.

CRC colorectal cancer

AA advanced adenoma

NAA non-advanced adenoma

The Chi-square test was used to compare the data, and P-values < 0.05.

Stool DNA-NEUROG1 methylation detection in CRC

The methylation level of the NEUROG1gene was evaluated by collecting stool samples from 272 patients, including (104 CRC, 92 AA, 39 NAA) and 37 healthy cohorts. The clinical characteristics of these patients are listed in Table 3. The ages ranged from 18 to 87, with 153 males and 119 females. Next, NEUROG1 methylation in stool samples was tested in different cancer stages.

Table 3 Basic clinical characteristics of enrolled subjects.

Subgroup	Number	Age (range, years)	Age (medium, years)	Male	Female	
CRC	104	24–87	59	60 (57.69%)	44 (42.31%)	
AA	92	27–83	59	52 (56.52%)	40 (43.48%)	
NAA	39	40–81	58	28 (71.79%)	11 (28.21%)	
Healthy volunteers	37	18–83	56	13 (35.14%)	24 (64.86%)	
Notes.

CRC colorectal cancer

AA advanced adenoma

NAA non-advanced adenoma

There were no significant differences in age or sex between each subgroup.

In the 104 CRC patients, the NEUROG1 methylation level was 63.46%. To further evaluate the sensitivity of NEUROG1 methylation for different tumor classifications and positions, the methylation level was assessed in various CRC stages (I, II, III, IV), as shown in (Table 4).

Table 4 Detection rates of NEUROG1 methylation in stool samples of CRC.

	Number	Sensitivity	p-value	
CRC	104	63.46% (66)		
Classification:				
Stage I	26	65.38% (17)	0.201	
Stage II	24	62.50% (15)		
Stage III	27	70.37% (19)		
Stage IV	17	41.18% (7)		
Position:				
Proximal	27	48.15% (13)	0.067	
Distal	73	67.12% (49)		

In the 94 patients with known stages, the detection rate ranged from 41.18% to 70.37%, with no significant differences between each subgroup. The methylation rates of different tumor positions were explored among 100 patients. The methylation rates of proximal colorectal cancer was 48.15%, and for distal colorectal cancer, it was 67.12%. However, the methylation rates in CRC patients with distal CRC seemed higher, and the difference was not statistically significant, as shown in (Table 4).

Stool DNA-NEUROG1 methylation detection in AA and NAA

In 92 cases of AA, the methylation level was 47.83%. Further analysis was conducted to compare the rates in different subtypes of AA based on the tumor position and size. NEUROG1 methylation rates of proximal and distal AA were 36.67% and 57.41%, respectively (P > 0.05). Additionally, when the diameter of the adenoma was ≥3 cm, the detection rate increased significantly, reaching up to 61.90%, as shown in (Table 5). However, there were no significant differences between the different tumor sizes.

Table 5 Detection rates of NEUROG1 methylation in stool samples of AA.

	Number	Sensitivity	p-value	
AA	92	47.83% (44)		
Size (cm):				
≤1.9	44	43.18% (19)	0.338	
2.0–2.9	27	44.44% (12)		
≥3.0	21	61.90% (13)		
Position:			0.068	
Proximal	30	36.67% (11)		
Distal	54	57.41% (31)		

In the 39 NAA cases, 27 samples were negative, resulting in a detection specificity of 69.23%. In the 37 healthy volunteers, 31 samples were negative, resulting in a detection specificity of 83.78%. Hence, the NEUROG1 methylation detection sensitivity was 56.12% (110/196) in the positive group of 196 CRC and AA. By comparison, the detection specificity was 76.32% (58/76) in the cases of NAA and healthy people in the control group (Table 6).

Table 6 Test performance evaluation for NEUROG1 methylation in all participants.

Methylation	N	Methylation	
		n (Sensitivity)	PPV	PLR	
All	196	110 (56.12%,95% CI [49.08%–62.99]%)	85.94%95% CI [78.96%–91.47%])	2.37	
CRC	104	66 (63.46%, 95% CI [53.59%–72.56%])	
AA	92	44 (47.83%, 95% CI [37.48%–58.34%])	
		n (Specificity)	NPV	NLR	
All	76	58 (76.32%, 95% CI [65.45%–85.31%])	40.28%95% CI [32.31%–48.57%])	0.57	
NAA	39	27 (69.23%, 95% CI [53.35%-82.71%])	
Healthy control	37	31 (83.78%, 95% CI [68.97%–93.80%]	
Notes.

CRC colorectal cancer

AA advanced adenoma

NAA non-advanced adenoma

PPV Positive Predictive Value

NPV Negative Predictive Value

PLR Positive Likelihood Ratio

NLR Negative Likelihood Ratio

Figure 2 Result of ROC analysis evaluating 272 stool samples undergone NEUROG1 methylation.

Diagnostic performance of NEUROG1 methylation detection with stool samples

Receiver operating characteristic (ROC) curves were elaborated to evaluate the discriminatory capacity of NEUROG1, as shown in (Fig. 2). In our study, 272 stool samples were subjected to NEUROG1 methylation detection. Among those from NAA, healthy individuals were selected as controls and ROC curve analyses were performed. As shown in Fig. 2, the area under the curve (AUC) for the NEUROG1detection of CRC and AA was 0.699 (95% confidence interval (CI) [0.633–0.764]).

In the positive group (CRC and AA), the NEUROG1 methylation test demonstrated a sensitivity of 56.12% (95% CI [49.08%–62.99%]), a positive predictive value (PPV) of 85.94% (95% CI [78.96%–91.47%]), and a positive likelihood ratio (PLR) of 2.37 (Table 6).

In the control group (NAA and healthy volunteers), the test showed a specificity of 76.32% (95% CI [65.45%–85.31%]), a negative predictive value (NPV) of 40.28% (95% CI [32.31%–48.57%]), and a negative likelihood ratio (NLR) of 0.57 (Table 6).

NEUROG1 methylation rate was significantly different between CRC+AA group and NAA+ healthy volunteers group in stool samples, as shown in (Table 7).

Table 7 Comparation the rates of NEUROG1 methylation between CRC+AA group and NAA+ healthy control group in stool samples.

Subgroup	Number	Positive	Negative	
CRC+AA	196	110	86	
NAA+ Healthy control	76	18	58	
Notes.

CRC colorectal cancer

AA advanced adenoma

NAA non-advanced adenoma

The Chi-square test was used to compare the data, and P-values < 0.05.

Comparison of methylation detection between fecal DNA and serum DNA showed that the detection sensitivity and specificity in fecal DNA was better than in serum DNA.As shown in (Table 8). Therefore, NEUROG1 has the potential to serve as a marker for CRC and AA screening in stool samples.

Table 8 Comparison of methylation detection between fecal DNA and serum DNA.

	Sensitivity for CRC	Specificity for Healthy control	
Fecal DNA in this study	63.46%, 95% CI [53.59%–72.56%])	83.78% (68.97%–93.80%)	
Serum DNA Herbst et al., 2011	55.5%	81.3%	
	Sensitivity for CRC+AA	Specificity for NAA+ Healthy control	
Fecal DNA in this study	56.12%, 95% CI [49.08%–62.99%]	76.32%, 95% CI [65.45%–85.31%]	
Serum DNA Otero-Estévez	33.33%, 95% CI [21.4%–47.1%]	90.60%, 95% CI [87.5%–93.1%]	
Notes.

CRC colorectal cancer

AA advanced adenoma

NAA non-advanced adenoma

Discussion

Methylation in the promoter region of NEUROG1 was identified long ago, and applications of NEUROG1 methylation in blood DNA tests have been reported in CRC (Goel et al., 2010; Herbst et al., 2011; Li et al., 2018). Detection rate of NEUROG1 methylation for CRC in serum samples was 55.5%. The specificity was 81.3% for healthy control (Goel et al., 2010; Herbst et al., 2011; Li et al., 2018). However, studies of NEUROG1 methylation in the stool have not been reported. This study aimed to study the methylation of NEUROG1 in stool, tissues samples from patients with CRC, AA, and NAA. CRC development is a complex process of five steps, including normal intestinal epithelium, NAA, AA, adenocarcinoma, and finally, cancer metastasis (Siraj et al., 2014). This process is associated with a large number of oncogene and tumor suppressor gene disturbances, including mutations of APC, Kirsten rat sarcoma viral oncogene homolog (KRAS), and Tumor Protein P53 (TP53), microsatellite instability, and DNA methylation abnormalities (Okugawa, Grady & Goel, 2015). In patients with CRC, a proportion of tumor cells can be shed from the gut into the stool (Diehl et al., 2008). Therefore, the preliminary for CRC can be conducted by detecting the DNA shedding in stool samples. Previously, Sidransky and colleagues discovered the presence of the KRAS gene in stool samples from CRC patients in 1992 (Sidransky et al., 1992). Unfortunately, the sensitivity of the DNA mutation detection in stool samples was too low, and screening was difficult.

Many studies have reported the role of DNA methylation and suggested that DNA methylation detection could be a great approach for screening early-stage CRC (Barault et al., 2018). However, in a previous report, the sensitivity of DNA methylation detection for a single gene in the early stages of the disease was low (Zhao et al., 2020).

Various DNA methylation molecular targets could be used as biomarkers in CRC. For example, one study explored the role of Syndecan-2 (SDC2) methylation in CRC and found that SDC2 methylation could be detected in 81.1% (159/196) CRC and 58.2% (71/122) adenomas in stool samples (Niu et al., 2017). In addition, the tumor suppressor gene, tissue factor pathway inhibitor 2 (TFPI2), was reported to be methylated in CRC. The sensitivity of methylated TFPI2 in stool DNA of stage I-III CRC patients was 76% to 89%, with a specificity of 79% to 93% (Glöckner et al., 2009). Recently, one report implied that potassium voltage-gated Channel Subfamily Q Member 5 (KCNQ5) methylation and Chromosome 9 Open Reading Frame 50 (C9orf50) methylation in stool DNA could be possible biomarkers for CRC detection, with sensitivities of 77.3% and 85.9%, and specificities of 91.5% and 95.0%, respectively (Niu et al., 2017). In addition, one study explored the application of Phosphatase and Actin Regulator 3 (PHACTR3) methylation in CRC stool DNA, with a sensitivity of 55% (95% CI [33–75]) and specificity of 95%. In our study, we investigated the role of methylated NEUROG1 in the stool DNA of CRC patients and found the sensitivity was 63.46% and specificity was 76.32%. In addition, accumulating evidence has evaluated stool-based DNA methylation markers for CRC diagnosis. Although methylated genes, such as SDC2, SFRP2, and TFPI2, have been demonstrated to possess the capacity to detection of established cancers (Glöckner et al., 2009; Sun et al., 2019; Wang et al., 2020; Zhang et al., 2014), the goal of effective diagnosis has not been achieved yet. For instance, the most known methylation marker approved by the FDA is SEPT9. A prospective analysis of SEPT9 methylation detection showed a sensitivity of 48.2% for CRC and a sensitivity of only 11.2% for AA (Church et al., 2014). These findings have compelled us to consider evaluating multiple markers for a more precise diagnosis.

In this study, the detection rate of NEUROG1 methylation in AA FFPE tissue was significantly higher than that in CRC and NAA tissue. We speculate that NEUROG1gene methylation may occur in precancer or early-stage of cancer and participate in the process of tumor progression.

The overall NEUROG1 methylation detection sensitivity for CRC in stool samples was 63.46% (95% CI [53.59%–72.56%]). The detection rate of stage I-III was similar whereas the detection rate of stage IV was relatively low, which may be due to the small number of cases (17 cases). It is also possible that the gene becomes unmethylated due to several factors in stage IV. The sample size should be increased in further study.

In stool samples, the overall NEUROG1 methylation detection sensitivity for AA was 47.83%. There were no significant differences between the different tumor sizes.

Currently, the research on NEUROG1 methylation in CRC mainly focuses on serum to detect the early stage of CRC (Herbst et al., 2011). Studies of NEUROG1 methylation in the stool have not been reported. Our study initially clarified its detection performance and laid a certain foundation for the later development of detection methods using this gene or the combination of this gene with the other genes. This study also has some limitations. First, future studies should evaluate the diagnostic performance of stool DNA testing in large cohorts of NAA patients, healthy volunteers and patients with other upper gastrointestinal diseases, inflammatory bowel diseases and liver cancer patients. Second, most of the samples are from Fujian Provincia Hospital, which may limit the representativeness of the findings. Further studies with large sample size are needed to provide more reliable evidence for the clinical application of combined methylation detection.

Conclusions

In stool samples, NEUROG1 methylation detection sensitivity for CRC was 63.46%. The methylation level was 47.83% for AA. In the healthy volunteers, the test showed a specificity of 83.78%. Comparison of methylation detection between fecal DNA and serum DNA showed that the detection sensitivity and specificity in fecal DNA was better than in serum DNA. Therefore, NEUROG1 has the potential to serve as a marker for CRC and AA screening.

Supplemental Information

Supplemental Information 1 Raw data

Supplemental Information 2 Code for the data

Supplemental Information 3 Codebook

Additional Information and Declarations

Competing Interests

Author Contributions

Human Ethics

Data Availability

Tao Wang, Yan Lu, Zhaofen Ba, Xiaowei Cheng and Tiantian Zhang are employed by Jiangsu Microdiag Biomedical Technology Co., LTD. All other authors declare no competing interests.

Lijin Zhang analyzed the data, prepared figures and/or tables, authored or reviewed drafts of the article, and approved the final draft.

Aiping Lin analyzed the data, prepared figures and/or tables, authored or reviewed drafts of the article, and approved the final draft.

Jie Lin performed the experiments, authored or reviewed drafts of the article, and approved the final draft.

Juan Chen analyzed the data, prepared figures and/or tables, authored or reviewed drafts of the article, and approved the final draft.

Mengshi Chen performed the experiments, authored or reviewed drafts of the article, and approved the final draft.

Xunbin Yu performed the experiments, authored or reviewed drafts of the article, and approved the final draft.

Yijuan Wu performed the experiments, authored or reviewed drafts of the article, and approved the final draft.

Tao Wang conceived and designed the experiments, authored or reviewed drafts of the article, and approved the final draft.

Yan Lu analyzed the data, prepared figures and/or tables, authored or reviewed drafts of the article, and approved the final draft.

Zhaofen Ba analyzed the data, prepared figures and/or tables, authored or reviewed drafts of the article, and approved the final draft.

Xiaowei Cheng analyzed the data, prepared figures and/or tables, authored or reviewed drafts of the article, and approved the final draft.

Tiantian Zhang performed the experiments, authored or reviewed drafts of the article, and approved the final draft.

Minghong Chen conceived and designed the experiments, authored or reviewed drafts of the article, and approved the final draft.

The following information was supplied relating to ethical approvals (i.e., approving body and any reference numbers):

This study was approved by the Institutional Review Board of Fujian Provincial Hospital (K2019-11-027). Informed consent was obtained from all enrolled patients and healthy control subjects.

The following information was supplied regarding data availability:

The raw measurements are available in the Supplemental Files.

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
