# Peer review of "Evaluation of NEUROG1 methylation status in stool specimens in the advanced adenomas and colorectal cancer"

_PeerJ, doi:10.7717/peerj.19968_

## Round 0.1 · original submission · Major Revisions

The reviewers have provided valuable comments. In particular, the concerns of Reviewer 1 and 4 must be carefully addressed.

Reviewer 1 ·

Basic reporting

The author does a decent job to systematically study the methylation of NEUROG1 in stool from patients with CRC, AA, and NAA.
The reporting is fine- but not much is done here to support their claims. The author needs to dial down the tone of the paper to only what the experiments show, rather than claiming that this is the next big thing in CRC diagnosis.
Literature references seem ok.

Experimental design

1. Line 199: The authors need to clarify by what do they actually mean by not low in "The detection rates for CRC and NAA were not low, at 36% for CRC 24% for NAA". What is the comparison here?
2. What stage are these cancers?
3. What is the detection rate for NEUROG1 methylation in serum?
4. Table 5: methylation comparison from the serum are missing.
5. Line 239-242- please correct the spacings etc.
5. Tables: Please 95% confidence interval numbers for sensitivities, specificities, NPV, PPV etc.

Validity of the findings

The number of samples used for study are low than typical studies of these kinds.
How does this manuscript show that NEUROG1 methylation1 has the potential to be a biomarker for precancerous colon lesions? Data isn't there to prove this other than a hypothesis.

Additional comments

Overall- 2 things must be done (along with toning down the claims) to accept this manuscript.
1) Comparison of stool data to serum data; and
2) Increase the number of samples used here to solidify the confidence in the results of this manuscript.

·

Basic reporting

The writing could be improved.
The literature review provided adequate information.
All tables were clear, but Figure 1 requires additional details.
The results were not significant and deviated considerably from the hypothesis.

Experimental design

The study was original and aligned with the journal's aim and scope.
The research question was clearly defined, relevant, and addressed a recognized knowledge gap.

A thorough investigation was conducted, adhering to high technical and ethical standards.

However, the methods section requires more detail, particularly regarding the specific techniques used.

Validity of the findings

in the conclusion part, it needs to compare the artiche result with previous results in serum sample

Additional comments

Dear Author,
The research project was valuable for detecting the early stage of colon cancer in the stool sample. However, some issues were not clear.
1. You mentioned designing forward and reverse primers for ACTB gene; also, you mentioned prob without talking about its result part. What is the function of the ACTB gene?
2. After using the bisulfite conversion kit, all unmethylated cytosine is converted to uracil, and all sites of unmethylated cytosine are displayed as thymine in the resulting amplified sequence. There is no sequence of designed primers specific to the methylated or unmethylated CpG site for PCR amplification, remarking the site of the CpG.
3. Sequence of the promoter region: which software is used for detecting the gene's promoter region? Also, it is better to include the whole sequence of the promoter region.
4. Mention the specific region of the CpG site at the promoter region, the number of the upstream base pair, for example, -70 or 237
5. Function of this CpG site for gene transcription, if it's the site enhancer of insulator
6. How does methylation level contribute to transcription? Does methylation restrict the binding of the transcription factor to the promoter region, or does it enhance binding (there are transcription factors that bind to the methylated CpG, and others cannot bind to the CpG site?
7. You put the result part in Figure 1. Instead of that, it is better to put the number of cancer cases according to the stages

Reviewer 3 ·

Basic reporting

Overall, the paper is well written and easy to follow. References are sufficient and refer to the appropriate articles. The data used to calculate the results are made available.

Experimental design

The overall design of the experiments is solid, there are just some missing details.
1. How was the specific region of the NEUROG1 promoter chosen for evaluation? Was it based on the specific location defined in the Otero-EstEvez and Gallardo-Gomez paper, or for some other reason? The authors should be explicit about this.
2. The authors should describe how the threshold of 0.7848 determined. Was it set for some specific sensitivity analysis?
3. The authors use PPV and PLR as metrics in the results, but never describe what they are. They should be written out in full at least once in the manuscript or the table legend.

Validity of the findings

In the discussion, the authors state that "NEUROG1 methylation has a good sensitivity in the detection of CRC and precancer, especially in the detection of precancerous stage." However, it is unclear what set of results the authors are referring to. This does not seem like a valid statement for the stool-based results in which the AA has a sensitivity less than 50%, nor the FFPE, where the CRC only has a 36% detection rate. The authors should either provide a bit more justification to that statement, or clarify exactly which results they are referring to.

Reviewer 4 ·

Basic reporting

The authors assessed the performance of NEUROG1 methylation in the colorectal cancer auxiliary diagnosis. There are several problems:
1. The sample size (25 CRC, 25 non-advanced adenomas, 25 advanced adenomas) is too small.
2. How reliable were the stool samples? Have all samples passed the quality control?
3. The authors should add a complete workflow to show all analysis steps.
4. Many details were not clear. For example, in “the sample is positive if the fitting value was >0.7848”, how was 0.7848 determined?
5. There are many typos. For example, in “In the control group of NAA and healthy volunteers, the specificity value of NEUROG1 methylation detection was77.92%, NPV41.10%, NLR0.56 (Table 5).”, 0.56 should be percentile.

Experimental design

The sample size (25 CRC, 25 non-advanced adenomas, 25 advanced adenomas) is too small.

Validity of the findings

How reliable were the stool samples? Have all samples passed the quality control?

Additional comments

The authors should add a complete workflow to show all analysis steps.

Many details were not clear. For example, in “the sample is positive if the fitting value was >0.7848”, how was 0.7848 determined?

There are many typos. For example, in “In the control group of NAA and healthy volunteers, the specificity value of NEUROG1 methylation detection was77.92%, NPV41.10%, NLR0.56 (Table 5).”, 0.56 should be percentile.

---

## Round 0.2 · Major Revisions

Reviewer 1 still has some serious concerns. It is important to address these comprehensively for the manuscript to be suitable for publication.

Reviewer 3 ·

Basic reporting

In the introduction, the authors state that 'This study aimed to evaluate the performance of the NEUROG1 methylation test'. However, this manuscript and the previous ones that test NEUROG1 methylation all use slightly different methodologies for evaluating methylation status. Thus, this is far from a standard 'methylation test' and is more of an evaluation of the methylation status of NEUROG1 in evaluating CRC presence.

Experimental design

Two separate reviewers asked for the specific location of the amplified NEUROG1 promoter region being tested to be included in the text. The authors provided the location in the response letter, but still did not include it in the main text. Please include this so future studies can replicate the findings.

Validity of the findings

The authors claim that the samples must have an ACTB CT value less than 36 to pass quality control (Note: figure 1 now suggests that it's 30, but the text says 36). However, there are 2 samples in their raw data sheet out of the 274 that have ACTB CT values greater than 36, which should suggest that they do not pass QC and thus should not be included in the analysis. There are about 55 samples with an ACTB CT value > 30 (if figure 1 is to believed instead of the text).

My initial question about how the threshold of 0.7848 was determined has now gone from requiring a clarification to a possible not understanding of how it was done in the first place. In the initial text, the authors describe using a titration-type experiment with control DNA from cell lines. In response to my question about how the threshold was determined, the authors now state that they used the 274 samples that are tested using that threshold to determine that threshold using an ambiguously described "ROC analysis". If in face the testing data was used to determine the threshold, then the findings should not be considered valid.

Reviewer 4 ·

Basic reporting

The authors have answered my questions.

Experimental design

The authors have answered my questions.

Validity of the findings

The authors have answered my questions.

Additional comments

The authors have answered my questions.

---

## Round 0.3 · accepted · Accept

All issues indicated by the reviewers were addressed and the revised manuscript is acceptable now.

Reviewer 3 ·

Basic reporting

The authors have addressed my concerns.

Experimental design

The authors have addressed my concerns.

Validity of the findings

The authors have addressed my concerns.

Additional comments

small detail: Figure legends still show the incorrect number of samples (274 instead of 272).